# Systematic Review of Pharmacogenetics of ABC and SLC Transporter Genes in Acute Myeloid Leukemia

**DOI:** 10.3390/pharmaceutics14040878

**Published:** 2022-04-17

**Authors:** Juan Eduardo Megías-Vericat, David Martínez-Cuadrón, Antonio Solana-Altabella, José Luis Poveda, Pau Montesinos

**Affiliations:** 1Servicio de Farmacia, Área del Medicamento, Hospital Universitario y Politécnico La Fe, Avda. Fernando Abril Martorell 106, 46026 Valencia, Spain; megias_jua@gva.es (J.E.M.-V.); solana_ant@gva.es (A.S.-A.); poveda_josand@gva.es (J.L.P.); 2Servicio de Hematología y Hemoterapia, Hospital Universitario y Politécnico La Fe, Avda. Fernando Abril Martorell 106, 46026 Valencia, Spain; martinez_davcua@gva.es; 3Instituto de Investigación Sanitaria La Fe, Avda. Fernando Abril Martorell 106, 46026 Valencia, Spain

**Keywords:** *SLCO1B1*, *ABCB1*, *SLC29A1*, *ABCG2*, *ABCC1*, polymorphism, anthracyclines, cytarabine, acute myeloid leukemia

## Abstract

Antineoplastic uptake by blast cells in acute myeloid leukemia (AML) could be influenced by influx and efflux transporters, especially solute carriers (SLCs) and ATP-binding cassette family (ABC) pumps. Genetic variability in *SLC* and *ABC* could produce interindividual differences in clinical outcomes. A systematic review was performed to evaluate the influence of *SLC* and *ABC* polymorphisms and their combinations on efficacy and safety in AML cohorts. Anthracycline intake was especially influenced by *SLCO1B1* polymorphisms, associated with lower hepatic uptake, showing higher survival rates and toxicity in AML studies. The variant alleles of *ABCB1* were related to anthracycline intracellular accumulation, increasing complete remission, survival and toxicity. Similar findings have been suggested with *ABCC1* and *ABCG2* polymorphisms. Polymorphisms of *SLC29A1*, responsible for cytarabine uptake, demonstrated significant associations with survival and response in Asian populations. Promising results were observed with *SLC* and *ABC* combinations regarding anthracycline toxicities. Knowledge of the role of transporter pharmacogenetics could explain the differences observed in drug disposition in the blast. Further studies including novel targeted therapies should be performed to determine the influence of genetic variability to individualize chemotherapy schemes.

## 1. Introduction

Acute myeloid leukemia (AML) is a clinically and biologically heterogeneous hematologic malignant disease characterized by an excess of blast cells in bone marrow and blood. Approximately 60–80% of young AML patients achieve complete remission (CR) using conventional 3 + 7 schedules of anthracyclines and cytarabine, which might be followed by an allogeneic hematopoietic stem cell transplant (allo-HSCT) to prevent relapse [1,2]. Unfortunately, half of these patients finally relapsed or died from different causes, including: low efficacy eliminating the minimal residual disease, severe toxicity of chemotherapy, refractory disease. This interindividual variability of outcomes between AML patients could be related to their genetic variability [3,4].

Drug uptake by blast cells can be affected by different transporters, including influx and efflux transporters, especially solute carriers (SLCs) and ATP-binding cassette family (ABC) pumps, respectively [3,4]. Previous pharmacogenetic studies have suggested that single nucleotide polymorphisms (SNPs) of *SLC* and *ABC* transporters may play a promising role in drug exposure and have been associated with clinical response and toxicity [3,4,5,6,7]. However, the findings and the interpretation of these individual studies appear contradictory and inconclusive. Furthermore, for new targeted therapies, potential drug–drug interactions with P-glycoprotein (P-gp), breast cancer resistance protein (BCRP) and organic anion transporting polypeptides (OATP) were tested in preclinical studies, but the influence of SNPs in these transporters is unknown in these new therapies. We performed a systematic review of all the studies that have analyzed polymorphisms of membrane transporters in AML patients.

## 2. Materials and Methods

### Search Strategy and Selection of Studies

A systematic search was performed following the PRISMA guidelines by two independent reviewers (JEMV and ASA) [8]. Pubmed, EMBASE, the Cochrane Central Register, the Web of Science and the Database of Abstracts of Reviews of Effects (DARE) databases were searched without restrictions. In addition, the reference lists of important studies and reviews were hand searched. The reference lists of relevant reviews and studies were manually searched. The last literature search was conducted on 26 January 2022. This systematic review was included in the PROSPERO registry (ID 314292).

Similar keywords were used in different databases: (“ATP-binding cassette transporters” [MeSH Terms] or “organic anion transporters” [MeSH] or “organic cation transport proteins” [MeSH]) and “acute myeloid leukemia” [MeSH].

Studies that fulfilled the following criteria were included: (1) studies based on clinical data in AML patients (excluding preclinical and in vitro studies); (2) AML studies analyzing the associations between *ABC* and/or *SLC* polymorphisms and clinical response to chemotherapy; and (3) AML studies analyzing the impact on safety of *ABC* and/or *SLC* polymorphisms.

## 3. Results

Our systematic search obtained 569 citations from databases and journals and 21 records were identified through other sources (Figure 1). Of the 44 citations selected for full reading, 37 fulfilled the inclusion criteria and were included. The agreement in the study selection between the reviewers was excellent (kappa = 0.97).

### 3.1. Influx Transporters: SLC Family

The intake by blast cells and other tissues of the antineoplastics employed in AML therapy and other xenobiotics is mediated by SLC transporters, a family that includes more than 400 transporters. Different SLC transporters have been related to anthracycline uptake, especially the organic anion transporter polypeptide-1B1 (OATP1B1, encoded by *SLCO1B1*) and the organic cation transporter *SLC22A16* (Figure 2). However, cytarabine is mainly transported by human equilibrative nucleoside transporter (hENT1 and hENT2, encoded by the *SLC29A1* and *SCL29A2* genes; Figure 2), and in lower proportions by the human concentrative nucleoside transporters (hCNT3 encoded by the *SLC28A3* gene).

The OATP1B1 (*SLCO1B1*) is predominantly expressed in the liver and is involved in the hepatic uptake and plasma clearance of several organic anionic compounds, including anthracyclines and other drugs such as statins [9,10,11,12]. The most relevant *SLCO1B1* polymorphisms are 521T>C (rs4149056), 388A>G (rs2306283) and 597C>T (rs2291075), which are partially in linkage disequilibrium. The minor allele of rs4149056 has been consistently associated with a lower hepatic uptake and higher drug circulating concentrations, increasing the plasma levels and the risk of toxicity in tissues [10,11]. In AML studies (Table 1), the variant allele of *SLCO1B1* rs4149056 was associated with a higher liver toxicity in adult patients [5] and higher overall survival (OS) in AML children [13]. In a recent study, the wild-type TT genotype of this SNP was related to a higher induction death, probably associated with a higher idarubicin uptake in tissues and therefore a higher potential toxicity [14]. The previous study in AML pediatric patients also obtained a higher OS and event-free survival (EFS) in carriers of the variant allele of the *SLCO1B1* polymorphism (rs2291075), as well as those of the *SLCO1B1* haplotype *1A/*1A,*1B/*1B (rs2291075, rs4149056 and rs2306283) [13].

*SLC22A12* encodes a solute carrier that is mainly expressed in kidney and other tissues and is involved in urate–anion exchange [15]. Moreover, it is associated with the transport of different drugs, especially uricosurics (allopurinol and oxypurinol). The wild-type homozygote of *SLC22A12* rs11231825 showed a higher infusion-related reactions after gemtuzumab ozogamicin administration (Table 1) [5]. An association between the wild-type genotypes of different *SLC22A12*, *SLC25A37* and *SLC28A3* polymorphisms showed a lower disease-free survival (DFS), although these associations were lost after the correction for multiple testing (Table 1) [16].

*SLC22A16* encodes an organic cation transporter of L-carnitine, a metabolism cofactor related to different disease states. This carrier also imports several drugs, including anthracyclines. This transporter is constitutively expressed in the brain and kidney. *SLC22A16* is over-expressed in AML and is related to the growth and viability of the blast cells, providing a potential target for future AML therapies [17]. In breast cancer cohorts, variant alleles of *SLC22A16* (rs714368) were found to be related to higher exposure levels of doxorubicin and doxorubicinol [6] and dose delays by anthracycline toxicities (lower with rs714368, rs6907567, rs723685 and higher with rs12210538) [7]. In a recent AML study, associations were not observed between *SLC22A16* rs12210538 and rs714368 and response or safety outcomes (Table 1) [14].

The concentrative nucleoside transporter hCNT1 encoded by the *SLC28A1* gene has a substrate specificity for physiological pyrimidine nucleosides. Besides this function, hCNT1 has been implicated in tumor suppression. Various *SLC28A1* SNPs were analyzed in several AML studies [18,19,20] (Table 1). Carriers of the *SLC28A1* rs2242046 polymorphism showed a higher neutropenia [19], whereas studies with *SLC28A1* rs2290272 [18] and *SLC28A1* rs8025045 [20] did not find any clinical association. *SLC28A2* encodes a sodium-dependent selective transporter of purines expressed in the kidney and other tissues. Only a pediatric AML study found a lower OS and EFS with the wild-type genotype of *SLC28A2* rs10519020 [13].

hCNT3 (*SLC28A3* gene) is a sodium-dependent pyrimidine and purine nucleoside carrier expressed in the pancreas, trachea, bone marrow and mammary glands. hCNT3 is a minor cytarabine transporter compared to hENT1, and this carrier has been associated with the uptake of different anthracyclines [21]. In four pediatric cancer cohorts, the variant alleles of *SLC28A3* rs7853758 and rs885004 were correlated with cardiotoxicity associated with anthracyclines (doxorubicin and daunorubicin) [22,23,24,25], whereas this finding with *SLC28A3* rs7853758 was not reproduced in cohorts of breast cancer [26,27] or B-cell lymphoma [28]. *SLC28A3* rs7853758 and rs885004 SNPs are in high linkage disequilibrium and have been related to lower expression in different cell lines [29,30]. Only one study in AML patients has reported an association of *SLC28A3* rs11140500 with a lower DFS, but the significance disappeared after Bonferroni correction (Table 1) [16].

hENT1 (encoded by the *SLC29A1* gene) is responsible for up to 80% of cytarabine influx in blast cells. Schemes with high doses of cytarabine (2–3 g/m^2^ daily), used in consolidation or intensification therapy, can saturate the pump-mediated transport of hENT1 with concentrations >10 µmol/L and produce free diffusion into the cell [31,32]. Nevertheless, intracellular cytarabine concentrations obtained with induction therapy (200 mg/m^2^) are mediated by hENT1 [33]. Moreover, the intracellular influx is strongly correlated with the abundance of hENT1 in cell surface [34], so the bioavailability and clinical response depend on hENT1 expression [35]. In addition, *SLC29A1* expression can be affected by hypoxia inducible factor 1 (Hif-1) at the promoter or by the transcription factor peroxisome proliferator activated receptorα (PPARα) [36,37]. In AML, patients with a low *SLC29A1* mRNA expression had a significantly shorter DFS and OS in an adult cohort [38], but this had no influence in a pediatric AML cohort [39].

Two nonsynonymous and four synonymous polymorphisms were identified in a functional study of *SLC29A1*, but no influence in cytarabine uptake was measured [40]. In contrast, the haplotype of three *SLC29A1* polymorphisms (−1345C>G, −1050G>A and 706G>C) was correlated with higher mRNA expression [41]. Another study showed only a modest elevation in hENT1 gene expression with the variant −706G>C, but no influence on cytarabine toxicity in normal blood cells [42]. The minor alleles of *SLC29A1* polymorphisms only reach relevant frequencies in Asian populations, as is reflected in AML studies (Table 1). The variant A allele *SLC29A1* rs3734703 was associated with a lower OS and RFS alone [43] or combined with *TYMS* rs2612100 [44], but a higher CR was related to the A allele [20] and CC + AA genotypes [43]. The *SLC29A1* rs9394992 polymorphism was related to a lower CR [43], OS, DFS and mRNA expression, and a higher relapse rate (RR) [45], but no influence was found in another cohort [46]. Similarly, the variant allele of *SLC29A1* rs324148 (alone or in combination rs9394992) was associated with a lower OS, DFS and mRNA expression, and a higher RR [45], as well as a higher CR haplotype ht3 with rs3734703, rs9394992, rs693955, rs507964 and rs747199 but had no effect alone [43]. On the other hand, the *SLC29A1* rs693955 polymorphism was correlated to a lower time to relapse and neutropenia recovery [4].

### 3.2. Efflux Transporters: ABC Family

The *ABC* family of transporters includes several efflux pumps involved in the active efflux of drugs and xenobiotics from inside the cells with a potential increase in drug resistance [47]. The effect of these pumps is well-known in anthracycline disposition in blast cells and tissues, highlighting *ABCB1*, *ABCC1*, *ABCC3* and *ABCG2* (Figure 2) [47,48]. In addition, cytarabine uptake is influenced by two members of the “multidrug resistance-associated protein” (MRP) family, MRP7 and MRP8 (encoded by *ABCC10* and *ABCC11* genes), which have been related to deoxynucleotide efflux (Figure 2) [49,50].

The P-glycoprotein (P-gp), encoded by the *ABCB1* gene, is the most studied efflux pump of the *ABC* family. The pharmacogenetics of *ABCB1* have been widely analyzed in AML patients (Table 2), especially *ABCB1* 3435C>T (rs1045642), 2677G>A/T (rs2032582) and 1236C>T (rs1128503) polymorphisms [13,16,19,20,51,52,53,54,55,56,57,58,59,60,61,62,63,64,65,66,67,68,69]. An in vitro study associated the P-gp expression with a lower intracellular daunorubicin accumulation [70]. The pharmacokinetics of daunorubicin and its metabolite daunorubicinol were not affected by *ABCB1* polymorphisms, nor was mRNA expression in an Indian AML cohort [69]. However, previous studies in breast cancer have shown a higher doxorubicin clearance and lower peak levels of doxorubicinol with the wild-type haplotype of *ABCB1* [47].

Lower pump function was related to the variant alleles of *ABCB1*, favoring anthracycline intracellular accumulation with a higher potential efficacy and toxicity [61,71,72], but some studies did not reproduce this effect [51,54,58]. Following this hypothesis, better responses (higher CR and survival rates) has been reported in AML cohorts with different *ABCB1* polymorphisms [51,57,59,61,63,65,66,67,68], whereas in other studies, these SNPs showed no influence or a worse response [20,53,54,55,56,58,62,64] (Table 2). This finding of a higher CR and OS with variant alleles of *ABCB1* 3435C>T, 2677G>A/T and 1236C>T was reproduced in two meta-analyses [73,74]. The study of Rafiee et al. showed an association between these 3 *ABCB1* SNPs and a higher EFS and DFS and a lower relapse rate on gemtuzumab ozogamicin, highlighting the role of P-gp in calicheamicin efflux [64].

The toxicity of anthracyclines has only been evaluated in four AML studies, showing no associations in two studies [54,56] and relevant anthracycline related-toxicities in two studies [60,62]. He et al. found higher nausea and vomiting grades (3/4) with wild-type genotypes of *ABCB1* 3435C>T and 2677G>A/T (alone and in haplotype) in an Asian cohort [60]. On the other hand, in a Caucasian cohort, the variant alleles of *ABCB1* 3435C>T, 2677G>A/T and 1236C>T and their haplotypes were associated with higher organ toxicities (renal, hepatic and neutropenia), as well as with higher induction death [62]. In other malignancies, *ABCB1* SNPs were correlated with higher cardiotoxicity [22,23,75,76], but this was not reproduced in these AML studies [54,56,60,62], nor in a large study analyzing the potential correlation between *ABCB1* polymorphisms and the left ventricular ejection fraction (LVEF) [77].

*ABCB11* encodes a canalicular transporter of bile salts also called the “bile salt export pump” (BSEP) which has been associated with the efflux of some anticancer drugs in liver cells. The *ABCB11* rs4668115 and *ABCB4* rs2302387 polymorphisms reduced transporter expression and were found to be related to ≥grade 3 transaminitis after anthracycline infusion (mithramycin) in patients with refractory thoracic malignancies [78]. The wild-type genotype of *ABCB11* rs4668115 was correlated with a lower OS and EFS in AML patients (Table 2) [13].

*ABCC1* encodes the MRP1 pump, which mediates the export of organic anions and drugs from the cytoplasm, including methotrexate, antivirals and anthracyclines. The function of this pump confers resistance to anticancer drugs by decreasing their accumulation in cells and by mediating ATP- and GSH-dependent drug export [79]. Pharmacokinetic in vitro studies have shown decreased transport and higher maximum velocity (Vmax) of doxorubicin disposition with *ABCC1* (rs60782127) [80], whereas MRP expression reduced the intracellular daunorubicin accumulation [70]. Previous studies in other cancers have associated *ABCC1* (rs3743527, rs246221, rs4148350) with higher cardiotoxicity [22,23,27,81]. A small cohort performed in an Arab population correlated the expression of 4 *ABCC1* SNPs with a lower CR, drug sensitivity and relapsed/refractory disease in acute leukemia (Table 2) [82]. Subsequently, several AML studies analyzed the role of different *ABCC1* genotypes in clinical outcomes and safety (Table 2) [20,56,62,83]. Despite the fact that the association between cardiotoxicity and *ABCC1* polymorphisms was not reproduced in AML [20,56,62,83], *ABCC1* rs4148350 was related to hepatotoxicity [62], *ABCC1* rs212090 with gastrointestinal toxicity and rs212091 and rs3743527 with myelosuppression [20]. In addition, the *ABCC1* rs212090 and rs3743527 variant alleles showed lower survival rates, whereas *ABCC1* rs129081 increased OS and DFS [83].

*ABCC2* expresses MPR2, an export pump localized to the apical membrane of polarized cells, especially those hepatocytes with functions in biliary transport. This protein appears to contribute to the drug resistance of different anticancer drugs including anthracyclines [84]. Polymorphisms of *ABCC2* have been correlated with anthracycline toxicities in other malignancies: cardiotoxicity in non-Hodgkin lymphoma (rs45511401) [26], in survivors of HSCT (rs8187710) [85] and in pediatric cancer (rs4148350) [22], febrile neutropenia in breast cancer (rs4148350) [27] and leucopenia in osteosarcoma (17222723) [86]. In AML patients, only one cohort has analyzed *ABCC2* rs8187710, without any significant influence in response or toxicity [62].

*ABCC3* encodes a protein that may play a role in biliary transport and the intestinal excretion of organic anions, which is also related to drug efflux. The expression of *ABCC3* was found to be significantly higher in AML patients resistant to daunorubicin [87]. Clinical studies in AML cohorts corroborated this finding with *ABCC3* polymorphisms (Table 2) [16,18,88]. A lower DFS was reported with variant alleles of *ABCC3* polymorphisms (rs4148405, rs1989983, rs2301835, rs8079740), whereas other *ABCC3* (rs2277624, rs757420) SNPs showed a higher DFS [16]. A similarly higher OS was observed with the variant allele of *ABCC3*, rs4793665 [18]. A recent cohort reproduced the previous findings of lower OS rates with the minor allele of *ABCC3*, rs4148405 [88].

The *ABCC10* and *ABCC11* genes encode the MRP7 and MRP8 pumps which can efflux cytarabine in blast cells [49,50]. Unfortunately, we have not found any studies regarding the genetic variability of *ABCC10* and *ABCC11* in AML populations. Sorafenib, an FLT3 inhibitor employed in AML, produces the inhibition of *ABC* pumps, avoiding the efflux of cytarabine by MRP7 and MRP8 pumps and thereby increasing the cytarabine-sensitivity of blast cells [89,90].

The *ABCG2* gene expresses the “breast cancer resistant protein” (BCRP), a well-known *ABC* pump responsible for anthracycline efflux [91]. BCRP is localized in the cell membranes of epithelial cells of the small intestine, liver, kidney, brain and placenta [92]. In AML, an overexpression of *ABCG2* was observed in 33% of blast cells and this BCRP expression correlated with a worse prognosis and lower OS [93,94,95,96]. The two most common *ABCG2* SNPs are rs2231137 and rs2231142, and the minor alleles of these SNPs are related to a reduced level of BCRP expression [92]. No influence in anthracycline pharmacokinetics was reported with *ABCG2* in an AML cohort with daunorubicin (rs2231137, rs2231142, rs769188) [69] or a breast cancer cohort with doxorubicin (rs2231142) [47]. Several studies have described the impact of *ABCG2* genotypes in AML (Table 2) [18,56,62,97,98]. Contradictory results were observed with *ABCG2* rs2231137, showing a lower OS and lower risk of toxicities ≥ grade three with the GG wild-type genotype in a Caucasian cohort [56], but a higher OS and DFS in a mixed AML/ALL Asian cohort [97] and no influence in a Caucasian cohort [62]. On the other hand, three different cohorts reproduced an increase in OS in wild-type *ABCG2* rs2231142 carriers [56,97,98] and cardiac and lung toxicities were associated with the variant allele in another study [62]. Similar OS and DFS increases were obtained with the wild-type genotype of *ABCG2* rs2231149, as well as with its haplotype with the *ABCG2* rs2231137 and rs2231142 polymorphisms [97]. No effect in LVEF was observed with 16 different *ABCG2* polymorphisms in a large study [77].

### 3.3. SNP-SNP Combinations of Transporters

Most of the included pharmacogenetic studies employed the candidate genes approach based on the pharmacologic pathway of the drugs. The drug intake depends on the combination of input and output transporters, but only a few studies analyzed the genetic variability of both types of carriers together. A recent study explored the combination of *SLC* wild-type genotypes (functional *SLCO1B1* and/or *SLC22A16*), ensuring the anthracycline uptake in cells, with the variant genotypes of *ABC* pumps (defective expression of *ABCB1*, *ABCC1*, *ABCC2* or *ABCG2*), avoiding anthracycline expulsion [14]. Several novel findings were reported with the combinations of *ABCB1* and *SLC* polymorphisms, including higher hepatic and renal toxicities, mucositis and neutropenia, as well as a higher incidence of induction death (Table 3). All of these are probably associated with a higher intracellular idarubicin accumulation and have been previously reported with *ABCB1* SNPs [62]. In addition, the combination of the *SLC22A16* rs714368 wild-type genotype with the variant allele of *ABCG2* rs2231142 was related to a higher cardiac toxicity (Table 3), reproducing the previous association [62]. On the other hand, no associations were found with *ABCC1* rs4148350 and *ABCC2* rs8187710 SNPs combined with *SLCO1B1/SLC22A16* wild-type genotypes. Combinations of *SLCO1B1* and *ABC* polymorphisms were also described with irinotecan [99,100] and statins [101,102]. Regarding cytarabine intake, two different studies analyzed the combined influence of SNPs in *SLC29A1* with genes of the main enzymes of the cytarabine pathway (*DCK*, *CDA*, etc.) [43,44], but the combination with *ABC* pumps was not explored.

## 4. Conclusions

Transporters of the *SLC* and *ABC* families play crucial roles in the absorption, disposition and elimination of antineoplastic drugs. In AML, the expression of these transporters has been proposed as one of the main drug resistance mechanisms and has been widely studied for standard chemotherapy 3 + 7 schedules based on anthracyclines and cytarabine. However, the impact of genetic variability in the *SLC* and *ABC* genes remains controversial. This review aims is to demonstrate that polymorphisms in transporter genes may have a potential impact on the clinical outcomes of AML therapy.

Despite this, only a few studies have analyzed the role of *SLC* carriers in AML therapy; promising findings were obtained with polymorphisms in the *SLCO1B1* and *SLC29A1* genes. Variant alleles of *SLCO1B1* were correlated with a lower function, decreasing anthracycline hepatic uptake and metabolism [10,11] and showed higher survival rates and toxicity in AML studies [5,13,14]. Polymorphisms of *SLC29A1*, responsible for cytarabine uptake, showed a relevant impact on CR and survival rates, especially in Asian populations [20,42,43,44,45].

Meanwhile, the variant alleles of *ABCB1* have been widely studied in AML, demonstrating a clear association with lower pump function, as well as higher CR and survival rates in meta-analyses [73,74]. The influence of *ABCB1* polymorphisms in anthracycline-related toxicities remains more controversial in AML, with scarce relevant findings [60,62] and without evidence of higher cardiotoxicity unlike studies in other malignancies [22,23,75,76]. Encouraging relationships were discovered in AML studies with *ABCC1* [20,62,82,83] and *ABCG2* polymorphisms [56,62,97,98].

SNP–SNP combinations of transporters could play a crucial role in characterizing the anthracycline pathway, which involves complex pharmacokinetic and pharmacodynamic mechanisms, although this was only evaluated in a Caucasian AML cohort [14]. In addition, it has been hypothesized that SNP–SNP combinations could increase the power of detection of significant associations where individual SNPs of *SLC* or *ABC* genes only demonstrate a minor effect that could be affected by their combination [103]. Combinations of transporters with other relevant SNPs such as enzymes have been explored in previous studies in AML with cytarabine [43,44].

The influence of *ABC* pumps in anthracycline pharmacokinetics has been suggested in vitro [70,80] and studies in other cancers [47], but a population pharmacokinetic study performed in AML failed to reproduce these findings with *ABCB1* and *ABCG2* polymorphisms [68]. Furthermore, the AML studies included did not analyze the influence of transporter SNPs together in drug pharmacokinetic levels and clinical response. In this line, a study in AML demonstrated a correlation between cytarabine plasma level and *CDA* genotype, the main enzyme responsible for liver metabolism of cytarabine [104]. In chronic myeloid leukemia, a relevant decrease in imatinib clearance was associated with variant alleles of *ABCB1* and *SLCO1B3* [105]. Similarly, in acute lymphoblastic leukemia, the *SLCO1B1* 521T>C SNP reduced methotrexate clearance [106]. Previous reviews focused on the impact of *ABC* and *SLC* SNPs in drug bioavailability have found the same limited evidence of PK studies in the AML context [47,107,108].

The influence of genetic variability in AML therapy has been previously analyzed by other authors, especially focused of the main SNPs of the cytarabine and anthracycline metabolic pathways [3,4,109,110] or only in SNPs of transporter genes [47,107,108,111]. Pinto et al. [112] recently performed a systematic review of the general state of pharmacogenetics in AML including, as a novelty, polymorphisms with a potential impact in new targeted therapies (e.g., FLT3 inhibitors, GO, hypomethylating agents and IDH inhibitors). On the other hand, our review centers on evaluating the influence of polymorphisms in transporter genes (*SLC* and *ABC* and their combinations) in AML studies, which was briefly explained in this recent review [112].

Most of the reported pharmacogenetic studies were performed in patients treated with a standard 3 + 7 scheme with a candidate genes approach. The importance of pharmacogenetics for the multiple new drugs recently approved for AML treatment remains unknown. Although these therapies are more tolerable than classical antineoplastics, potential drug–drug interactions involving P-gp, BCRP and OATP transporters have been described [113]. The genetic variability of *SLC* and *ABC* genes should be analyzed in further studies involving these novel therapies. In this line, a higher response to gemtuzumab ozogamicin was reported with the variant alleles of *ABCB1* in a pediatric cohort [63], but no influence was observed in adult AML patients treated with gemtuzumab ozogamicin and decitabine [64].

In conclusion, pharmacogenetic studies based on candidate genes have reported relevant associations between SNPs in transporters (*SLC* and *ABC*) with AML outcomes and safety profiles. Unfortunately, most of these studies were observational and involved retrospective cohorts, and only anecdotally were these transporter genes analyzed together with metabolic enzymes, molecular targets and DNA repair genes. In the future, randomized clinical trials on larger populations including those of different age, ethnic and therapy groups should be developed in order to validate the clinical benefit of pharmacogenetics in AML patients.

## Figures and Tables

**Figure 1 pharmaceutics-14-00878-f001:**
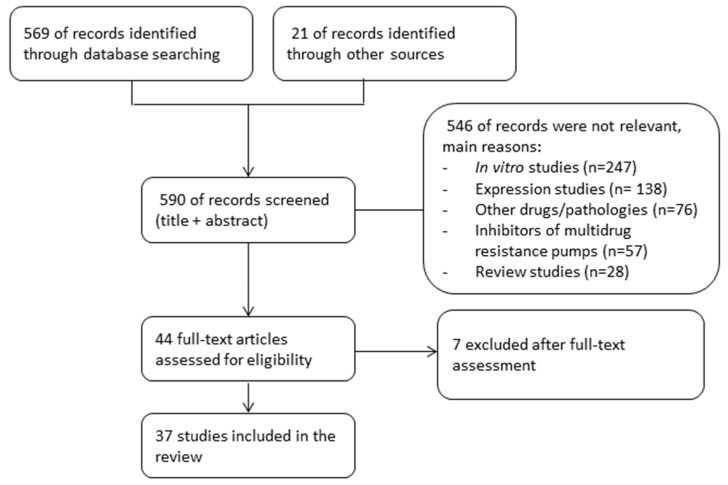
Summary of evidence search and selection.

**Figure 2 pharmaceutics-14-00878-f002:**
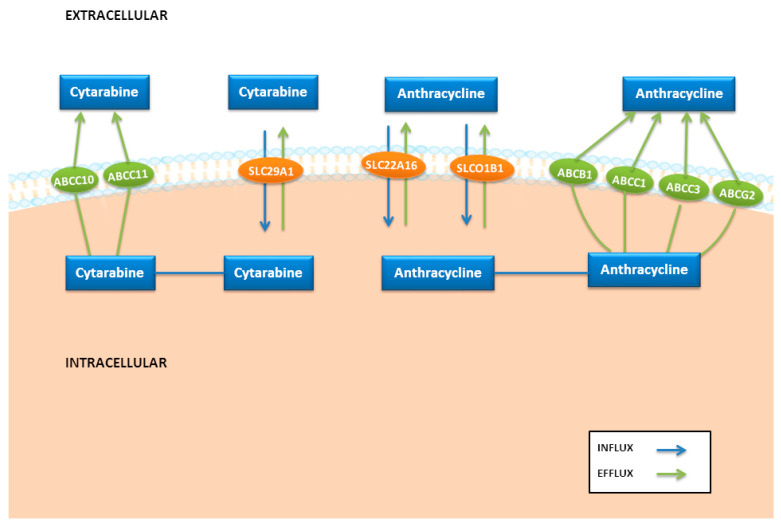
Key candidate genes involved in drug transport in acute myeloid leukemia.

**Table 1 pharmaceutics-14-00878-t001:** Characteristics of the studies included in the systematic review for influx transporters.

SNP	Study	n	Age (Range)	Ethnia (Country)	HWE	LMA Status (%)	Chemotherapy Scheme	Clinical Outcomes
** *SLCO1B1* **								
T521C rs4149056	Iacobucci et al., 2012 [5]	94	51 (19–65)	Caucasian (Italy)	Yes	De novo (80.9%) Secondary (19.1%)	Ara C + IDA + FLUDA + GO	- CR: no influence - Toxicity: CC/CT ↑liver toxicity
	Drenberg et al., 2016 [13] ^1^	164	9.1 (0–21)	White (70%) Black (20%) Others (10%)	Yes	De novo	Ara C + DAUNO + ETOP + MIT	- OS: TT ↓OS (p: 0.05) - EFS: no influence - Toxicity: no influence Haplotype *1A/*1A,*1B/*1B (rs2291075, rs4149056 & rs2306283): ↓OS y ↓EFS
	Megías-Vericat et al., 2021 [14]	225	52.5 (16–78)	Caucasian	Yes	De novo	Ara C + IDA	- CR: no influence - Induction death: TT ↑induction death (p: 0.049) - Toxicity: no influence
597C>T rs2291075	Drenberg et al., 2016 [13] ^1^	164	9.1 (0–21)	White (70%) Black (20%) Others (10%)	Yes	De novo	Ara C + DAUNO + ETOP + MIT	- OS: CC ↓OS (p: 0.012) - EFS: CC ↓EFS (p: 0.006) - Toxicity: no influence Haplotype *1A/*1A,*1B/*1B (rs2291075, rs4149056 & rs2306283): ↓OS y ↓EFS
388A>G rs2306283	Drenberg et al., 2016 [13] ^1^	164	9.1 (0–21)	White (70%) Black (20%) Others (10%)	Yes	De novo	Ara C + DAUNO + ETOP + MIT	- OS: no influence - EFS: no influence - Toxicity: no influence Haplotype *1A/*1A,*1B/*1B (rs2291075, rs4149056 & rs2306283): ↓OS y ↓EFS
** *SLC22A12* **								
T1246C rs11231825	Iacobucci et al., 2012 [5]	94	51 (19–65)	Caucasian (Italy)	Yes	De novo (80.9%) Secondary (19.1%)	Ara C + IDA + FLUDA + GO	- CR: no influence - Toxicity: TT/CT ↑fever reaction (associated with GO administration)
rs528211 (G>A)	Yee et al., 2013 [16] ^2^	154	NR	Caucasian (Europe)	NR	NR	Ara C + ETOP + BUSUL (pre-TX)	- DFS (preTX): GG ↓DFS (p: 0.0048). No influence in non-Caucasian cohort
rs2360872 (C>T)	Yee et al., 2013 [16] ^2^	154	NR	Caucasian (Europe)	NR	NR	Ara C + ETOP + BUSUL (pre-TX)	- DFS (preTX): CC ↓DFS (p: 0.0048). No influence in non-Caucasian cohort
rs505802 (A>G)	Yee et al., 2013 [16] ^2^	154	NR	Caucasian (Europe)	NR	NR	Ara C + ETOP + BUSUL (pre-TX)	- DFS (preTX): AA ↓DFS (p: 0.0048). No influence in non-Caucasian cohort
rs524023 (G>A)	Yee et al., 2013 [16] ^2^	154	NR	Caucasian (Europe)	NR	NR	Ara C + ETOP + BUSUL (pre-TX)	- DFS (preTX): GG ↓DFS (p: 0.0048). No influence in non-Caucasian cohort
rs9734313 (T>C)	Yee et al., 2013 [16] ^2^	154	NR	Caucasian (Europe)	NR	NR	Ara C + ETOP + BUSUL (pre-TX)	- DFS (preTX): TT ↓DFS (p: 0.0048). No influence in non-Caucasian cohort
rs11231825 (C>T)	Yee et al., 2013 [16] ^2^	154	NR	Caucasian (Europe)	NR	NR	Ara C + ETOP + BUSUL (pre-TX)	- DFS (preTX): CC ↓DFS (p: 0.0048). No influence in non-Caucasian cohort
rs11606370 (A>C)	Yee et al., 2013 [16] ^2^	154	NR	Caucasian (Europe)	NR	NR	Ara C + ETOP + BUSUL (pre-TX)	- DFS (preTX): AA ↓DFS (p: 0.005). No influence in non-Caucasian cohort
rs893006 (T>G)	Yee et al., 2013 [16] ^2^	154	NR	Caucasian (Europe)	NR	NR	Ara C + ETOP + BUSUL (pre-TX)	- DFS (preTX): TT ↓DFS (p: 0.0055). No influence in non-Caucasian cohort
** *SLC22A16* **								
rs12210538 1226A>G	Megías-Vericat et al., 2021 [14]	225	52.5 (16–78)	Caucasian (Spain)	Yes	De novo	Ara C + IDA	- CR and induction death: no influence - Toxicity: no influence
rs714368 146A>G	Megías-Vericat et al., 2021 [14]	225	52.5 (16–78)	Caucasian (Spain)	Yes	De novo	Ara C + IDA	- CR and induction death: no influence - Toxicity: no influence
** *SLC25A37* **								
rs7818607 (C>A)	Yee et al., 2013 [16] ^2^	154	NR	Caucasian (Europe)	NR	NR	Ara C + ETOP + BUSUL (pre-TX)	- DFS (preTX): AA ↓DFS (p: 0.0057). No influence in non-Caucasian cohort
rs8534 (C>T)	Yee et al., 2013 [16] ^2^	154	NR	Caucasian (Europe)	NR	NR	Ara C + ETOP + BUSUL (pre-TX)	- DFS (preTX): TT ↓DFS (p: 0.0067). No influence in non-Caucasian cohort
**CNT1** **(*SLC28A1*)**								
G565A rs2290272	Müller et al., 2008 [18]	139	46.3 (15–86)	Jews (61.2%) Arabs (38.8%)	Yes	De novo	Ara C + ANT ± FLUDA ± MIT	- OS (TX censured): no influence
C1561T rs2242046	Seeringer et al., 2009 [19] ^3^	322	<60	Caucasian (Germany)	NR	NR (normal cytogenetic status)	Ara C + IDA + ETOP	- Toxicity (hematologic): allele T reduced neutrophils and increased reconstitution time of total white blood cells
rs8025045 (G>T)	Cao et al., 2017 [20]	206	67.2 (22–98)	Asian (China)	Yes	De novo	Ara C + ANT	- CR: no influence - OS: no influence - RFS: no influence - Toxicity: no influence
** *SLC28A2* **								
rs10519020 (G>C)	Drenberg et al., 2016 [13] ^1^	164	9.1 (0–21)	White (70%) Black (20%) Others (10%)	Yes	De novo	Ara C + DAUNO + ETOP + MIT	- OS: GG ↓OS (p: 0.002) - EFS: GG ↓EFS (p: 0.001)
** *SLC28A3* **								
rs11140500 (C>T)	Yee et al., 2013 [16] ^2^	154	NR	Caucasian (Europe)	NR	NR	Ara C + ETOP + BUSUL (pre-TX)	- DFS (preTX): TT ↓DFS (p: 0.00018). No influence in non-Caucasian cohort
**hENT1** **(*SLC29A1*)**								
C469A rs3734703	Kim et al., 2013 [44] ^4^	97	50 (16–76)	Asian (South Korea)	Yes	De novo	Ara C + IDA	- CR, OS, RFS: no influence individually - OS, RFS: AA/AC combined with TYMS AA genotype (rs2612100) ↓OS and RFS (OS loses statistically significant after multivariable analysis) - Toxicity (hematologic): no influence
	Kim et al., 2016 [43]	103	50.4 (16–76)	Asian (South Korea)	Yes	De novo	Ara C + IDA	- CR: A allele ↑CR (p: 0.008; p Bonferroni:0.04) and haplotype ht3 (p: 0.01)
	Cao et al., 2017 [20]	206	67.2 (22–98)	Asian (China)	Yes	De novo	Ara C + ANT	- CR: CC+AA ↑CR (p: 0.027) - OS: AA+CA ↓OS (p: 0.006) - RFS: AA+CA ↓RFS (p: 0.003) - Toxicity: no influence
C>T rs9394992	Wan et al., 2014 [45]	100	43 (17–76)	Asian (China)	Yes	De novo	Ara C + DAUNO or IDA	- RR: CC ↓RR (p: 0.0004) - OS: CC ↑OS against CT (p: 0.02) and TT (p: 0.005) - DFS: CC ↑DFS against CT (p: 0.03) and TT (p: 0.001) - mRNA expression: CC ↑expression (*p* < 0.01) - SNP-SNP interaction: CT/TT + CC (rs324148) ↓OS (*p* < 0.001) and ↓DFS (p: 0.005)
	Amaki et al., 2015 [46]	39	54 (23–71)	Asian (Japan)	Yes	De novo	Ara C + IDA or DAUNO (consolidation: Ara C high doses)	- OS: no influence. - TTR: no influence. - Hematologic toxicity: no influence.
	Kim et al., 2016 [43]	103	50.4 (16–76)	Asian (South Korea)	Yes	De novo	Ara C + IDA	- CR: T allele ↑CR (p: 0.02; p Bonferroni:NS) and haplotype ht3 (p: 0.01)
T>C rs324148	Wan et al., 2014 [45]	100	43 (17–76)	Asian (China)	Yes	De novo	Ara C + DAUNO or IDA	- RR: CC ↑RR (p: 0.04) - OS: CC ↓OS against CT/TT (p: 0.0001) - DFS: CC ↓DFS against CT/TT (p: 0.0001) - mRNA expression: TT ↑expression (*p* < 0.01) - SNP-SNP interaction: CC+CT/TT (rs9394992) ↓OS (*p* < 0.001) and ↓DFS (p: 0.005)
	Kim et al., 2016 [43]	103	50.4 (16–76)	Asian (South Korea)	Yes	De novo	Ara C + IDA	- CR: no influence, ↑CR haplotype ht3 (p: 0.01)
A>C rs693955	Amaki et al., 2015 [46]	39	54 (23–71)	Asian (Japan)	Yes	De novo	Ara C + IDA or DAUNO (consolidation: Ara C high doses)	- OS: no influence. - TTR: CC ↓TTR (p: 0.00261; 0.0096 in multivariable analysis) - Hematologic toxicity: CC ↓neutropenia duration
	Kim et al., 2016 [43]	103	50.4 (16–76)	Asian (South Korea)	Yes	De novo	Ara C + IDA	- CR: no influence, ↑CR haplotype ht3 (p: 0.01)
rs507964 (A>C)	Kim et al., 2016 [43]	103	50.4 (16–76)	Asian (South Korea)	Yes	De novo	Ara C + IDA	- CR: C allele ↑CR (p: 0.03; p Bonferroni:NS) and haplotype ht3 (p: 0.01)
rs747199 (C>G)	Kim et al., 2016 [43]	103	50.4 (16–76)	Asian (South Korea)	Yes	De novo	Ara C + IDA	- CR: G allele ↑CR (p: 0.02; p Bonferroni:NS) and haplotype ht3 (p: 0.01)

Abbreviations: AMSA: amsacrine; ANT: anthracycline; BUSUL: busulfan; CR: complete remission; DAUNO: daunorubicin; DFS: disease-free survival; EFS: event-free survival; ETOP: etoposide; FLUDA: fludarabine; GO: gemtuzumab–ozogamicin; HWE: Hardy–Weinberg equilibrium; IDA: idarubicin; MIT: mitoxantrone; NR: not reported; OS: overall survival; RFS: relapse-free survival; RR: rate of relapse; TX: hematologic transplant. ^1^—This study [13] analyzed 1936 SNPs of 225 genes with a multi-SNP-based approach (including ABC and SLC transporters). Only SNPs with significant results were cited. ^2^—This study [16] analyzed 1659 SNPs of 42 genes with multi-SNP based approach. Only SNPs with significant results were cited. ^3^—This study [19] included SNPs of genes potentially involved in the response to Ara C (hCNT1, hENT1, hENT2, DCK, CDA), but only specified the SNPs with significant effect. ^4^—This study [44] included 139 SNPs of 10 genes potentially involved in the response to Ara C, but only specified the SNPs with significant effect.

**Table 2 pharmaceutics-14-00878-t002:** Characteristics of the studies included in the systematic review for polymorphisms of the ABC transporter family.

SNP	Study	n	Age (Range)	Ethnia (Country)	HWE	LMA Status (%)	Chemotherapy Scheme	Clinical Outcomes
** *ABCB1* **								
C3435T rs1045642	Illmer et al., 2002 [51]	405	53 (17–78)	Caucasian (Germany)	Yes	De novo	Ara C+ MIT + ETOP + AMSA	- CR: no influence. - OS and DFS at 4 years (TX censured): CC ↓OS (CC vs. CT *p* < 0.01, CC vs. CT/TT p: 0.05). - Haplotype with G2677T/A and C1236T: wild-type ↓OS and DFS at 4 years. - mRNA expression: CC ↓expression (*p* < 0.05)
	Kaya et al., 2005 [52]	28	36 (20–64)	Arabs (Turkey)	NR	NR	Ara C + ANT	- Drug sensitive/resistant: no differences (mixed with ALL cohort)
	Kim DH et al., 2006 [53]	81	39 (15–72)	Asian (South Korea)	Yes	De novo	Ara C + IDA	- CR: CC ↑CR (p: 0.05) - OS at 3 years (TX censured): no influence. - EFS at 3 years (TX censured): CC ↑EFS (p: 0.01) - Haplotype with G2677T/A: wild-type ↑CR and EFS at 3 years. - mRNA expression: CC ↓expression (p: 0.03)
	Van der Holt et al., 2006 [54] ^1^	150 (130)	67 (60–85)	Caucasian (Netherlands)	No	De novo: 79Secondary: 21	Ara C + DAUNO	- CR, OS, EFS, DFS at 5 years: no influence. - Expression and activity of P-gp: no influence
	Hur et al., 2008 [55]	200	44 (NR)	Asian (South Korea)	Yes	De novo	Ara C + ANT	- CR, OS, RFS and EFS at 5 years: no influence
	Hampras et al., 2010 [56]	261	61.5 (20–85)	Caucasian (86%) Others (14%) (USA)	Yes	De novo: 75Secondary: 25	Ara C + ANT	- OS (TX censured): no influence - Toxicity: no influence
	Green et al., 2012 [57]	100	63 (20–85)	Caucasian (Europe)	Yes	De novo (normal karyotype)	Ara C + ANT or MIT +/or Others	- OS at 4 years (TX censured): no influence
	Scheiner et al., 2012 [58] ^2^	109 (44)	34 (<1–86)	Others: White (69.7%) Non-white (30.3%)	No	De novo: 72.5Secondary: 18.3	Ara C + IDA	- OS at 5 years: no influence. - EFS at 5 years: CT ↑EFS (p: 0.001) - Expression and activity of P-gp: no influence
	Falk et al., 2014 [59] ^3^	201	59 (18–85)	Caucasian (Sweden)	Yes	De novo (normal karyotype)	Ara C + DAUNO or IDA ± ETOP +/or Others	- CR, OS, EFS: no influence (similar results in FLT3 wild-type subgroup).
	He et al., 2014 [60]	215	43.6 (14–57)	Asian (China)	Yes	De novo	Ara C (high doses)	- Toxicity: CC ↑acute nausea and vomiting grades 3/4 (p: 0.035, 0.010). In multivariable CC was a risk factor of vomiting (p: 0.016). - Haplotype with ABCB1 G2677T/A (rs2032582) CC/GG ↑acute nausea and vomiting grades ¾ (0.003; 0.026) and multivariable (0.003; 0.039)
	He et al., 2015 [61]	263	45.4 (14–58)	Asian (China)	Yes	De novo (intermediate cytogenetic risk)	Ara C + DAUNO ± MIT	- OS, RFS: TT ↑OS (p: 0.004), ↑RFS (p: 0.019) - Haplotype with G2677T/A and C1236T: TTT ↑OS (*p* < 0.001), ↑RFS (p: 0.005), both maintained in multivariable analysis (p: 0.001 and 0.009). - mRNA expression: TTT haplotype ↓mRNA expression than other genotypes (p: 0.004)
	Megías-Vericat et al., 2017 [62]	225	52.5 (16–78)	Caucasian	Yes	De novo	Ara C + IDA	- CR, induction death: no influence - Toxicity: TT genotype ↑renal toxicity (p: 0.008) - Haplotype C3435T, G2677T/A and C1236T: TTT ↑induction death (p: 0.020), ↑renal (p: 0.016) and hepatic (*p* < 0.001) toxicities.
	Rafiee et al., 2019 [63]	942	9.7 (0–30)	Caucasian (81%) Black (13%) Asian (5%) Others (1%)	Yes	De novo	Ara C + IDA + ETOP ± GO	- OS: in GO arm CT/TT trend to ↑OS at 5 years (p: 0.068) - EFS: in GO arm CT/TT↑EFS at 5 years (p: 0.022) - DFS: in GO arm CT/TT↑DFS at 5 years (p: 0.044) - RR: in GO arm CT/TT ↓RR at 5 years (p: 0.007) * These results were observed especially at standard risk group
	Short et al., 2020 [64]	104	68 (24–88)	Caucasian (86%) Black (13%)	NR	AML 82De novo: 43.9Secondary: 56.1	GO + DAC	- CR, ORR, CIR, OS, RFS: no influence
G2677T/A rs2032582	Van den Heuvel et al., 2001 [65]	30	34.6 (1–67)	Caucasian (Netherlands)	NR	Relapsed: 100	Ara C + ANT + Others	- OS after relapse at 3 years: GT ↑OS (p: 0.02) - RFS after relapse at 3 years: GT ↑RFS (p: 0.002)
	Illmer et al., 2002 [51]	405	53 (17–78)	Caucasian (Germany)	Yes	De novo	Ara C+ MIT + ETOP + AMSA	- CR: no influence. - OS and DFS at 4 years (TX censured): no influence. - Haplotype with C3435T and C1236T: wild-type ↓OS and DFS at 4 years. - mRNA expression: GG ↓expression (p: 0.05)
	Kaya et al., 2005 [52]	28	36 (20–64)	Arabs (Turkey)	NR	NR	Ara C + ANT	- Drug sensitive/resistant: no differences (mixed with ALL)
	Kim DH et al., 2006 [53]	81	39 (15–72)	Asian (South Korea)	Yes	De novo	Ara C + IDA	- CR: GG ↑CR (p: 0.04) - OS and EFS at 3 years (TX censured): no influence. - Haplotype with C3435T: wild-type ↑CR and EFS at 3 years. - mRNA expression: no influence.
	Van der Holt et al., 2006 [54] ^1^	150 (142)	67 (60–85)	Caucasian (Netherlands)	Yes	De novo: 79Secondary: 21	Ara C + DAUNO	- CR, OS, EFS, DFS at 5 years: no influence. - Expression and activity of P-gp: no influence
	Hampras et al., 2010 [56]	261	61.5 (20–85)	Caucasian (86%) Others (14%) (USA)	Yes	De novo: 75Secondary: 25	Ara C + ANT	- OS (TX censured): no influence - Toxicity: no influence
	Kim YK et al., 2010 [66]	94	38 (17–79)	Asian (South Korea)	NR	De novo (t (8,21) and inv (16))	Ara C + IDA +BH-AC	- CR, OS: no influence - RR: GG ↑RR (p: 0.031) - RFS: GG ↑RFS (p: 0.005)
	Green et al., 2012 [57]	100	63 (20–85)	Caucasian (Europe)	Yes	De novo (normal karyotype)	Ara C + ANT or MIT +/or Others	- OS at 4 years(TX censured): GG ↓OS (p: 0.02)
	Falk et al., 2014 [59] ^3^	201	59 (18–85)	Caucasian (Sweden)	Yes	De novo (normal karyotype)	Ara C + DAUNO or IDA ± ETOP +/or Others	- CR, OS, EFS: no influence - FLT3 wild-type subgroup: GG ↑OS (p: 0.039) against GT/TT genotypes.
	He et al., 2014 [60]	215	43.6 (14–57)	Asian (China)	Yes	De novo	Ara C (high doses)	- Toxicity: CC ↑acute nausea and vomiting grades 3/4 (p: 0.041, 0.038). Both lost in multivariable analyses. - Haplotype with ABCB1 G2677T/A (rs1045642) CC/GG ↑acute nausea and vomiting grades ¾ (0.003; 0.026) and multivariable (0.003; 0.039)
	He et al., 2015 [61]	263	45.4 (14–58)	Asian (China)	Yes	De novo (intermediate cytogenetic risk)	Ara C + DAUNO ± MIT	- OS, RFS: TT ↑OS (p: 0.017), ↑RFS (p: 0.033) - Haplotype with C3435T and C1236T: TTT ↑OS (*p* < 0.001), ↑RFS (p: 0.005), both maintained in multivariable analysis (p: 0.001 and 0.009) - mRNA expression: TTT haplotype ↓mRNA expression than other genotypes (p: 0.004)
	Megías-Vericat et al., 2017 [62]	225	52.5 (16–78)	Caucasian	Yes	De novo	Ara C + IDA	- CR, induction death: no influence - Toxicity: TT genotype ↑renal (p: 0.001), hepatic (p: 0.049) toxicities & ↑time to neutropenia recovery (p: 0.047) - Haplotype C3435T, G2677T/A and C1236T: TTT ↑induction death (p: 0.020), ↑renal (p: 0.016) and hepatic (*p* < 0.001) toxicities.
	Rafiee et al., 2019 [63]	942	9.7 (0–30)	Caucasian (81%) Black (13%) Asian (5%) Others (1%)	Yes	De novo	Ara C + IDA + ETOP ± GO	- OS: no influence - EFS: in GO arm GT/TT↑EFS at 5 years (p: 0.016) - DFS: in GO arm GT/TT ↑DFS at 5 years (p: 0.048) - RR: in GO arm GT/TT ↓RR at 5 years (p: 0.001)
C1236T rs1128503	Illmer et al., 2002 [51]	405	53 (17–78)	Caucasian (Germany)	Yes	De novo	Ara C+ MIT + ETOP + AMSA	- CR: no influence. - OS and DFS at 4 years (TX censured): no influence. - Haplotype with C3435T and G2677T/A: wild-type ↓OS and DFS at 4 years. - mRNA expression: no influence.
	Van der Holt et al., 2006 [54] ^1^	150 (115)	67 (60–85)	Caucasian (Netherlands)	Yes	De novo: 79Secondary: 21	Ara C + DAUNO	- CR, OS, EFS, DFS at 5 years: no influence. - Expression and activity of P-gp: no influence
	Hampras et al., 2010 [56]	261	61.5 (20–85)	Caucasian (86%) Others (14%) (USA)	Yes	De novo: 75Secondary: 25	Ara C + ANT	- OS (TX censured): no influence - Toxicity: no influence
	Kim YK et al., 2010 [66]	94	38 (17–79)	Asian (South Korea)	NR	De novo (t (8,21) and inv (16))	Ara C + IDA +BH-AC	- CR, RR, OS and RFS: no influence
	Green et al., 2012 [57]	100	63 (20–85)	Caucasian (Europe)	Yes	De novo (normal karyotype)	Ara C + ANT or MIT +/or Others	- OS at 4 years(TX censured): CC ↓OS (p: 0.03)
	Scheiner et al., 2012 [58] ^2^	109(44)	34 (<1–86)	Others: White (69.7%) Non-white (30,3%)	Yes	De novo: 72.5Secondary: 18.3	Ara C + IDA	- OS at 5 years: CC ↑OS (p: 0.04) - EFS at 5 years: CC ↑EFS (p: 0.007) - Expression and activity of P-gp: no influence
	Falk et al., 2014 [59] ^3^	201	59 (18–85)	Caucasian (Sweden)	Yes	De novo (normal karyotype)	Ara C + DAUNO or IDA ± ETOP +/or Others	- CR, OS, EFS: no influence - FLT3 wild-type subgroup: CC ↑OS (p: 0.017) against CT/TT genotypes.
	He et al., 2014 [60]	215	43.6 (14–57)	Asian (China)	No	De novo	Ara C (high doses)	- Toxicity: not analyzed (excluded by HWE)
	He et al., 2015 [61]	263	45.4 (14–58)	Asian (China)	Yes	De novo (intermediate cytogenetic risk)	Ara C + DAUNO ± MIT	- OS, RFS: TT ↑OS (p: 0.002), ↑RFS (p: 0.001) - Haplotype with C3435T and G2677T/A: TTT ↑OS (*p* < 0.001), ↑RFS (p: 0.005), both maintained in multivariable analysis (p: 0.001 and 0.009) - mRNA expression: TTT haplotype ↓mRNA expression than other genotypes (p: 0.004)
	Megías-Vericat et al., 2017 [62]	225	52.5 (16–78)	Caucasian	Yes	De novo	Ara C + IDA	- CR, induction death: no influence - Toxicity: TT genotype ↑renal (p: 0.001) and hepatic (p: 0.006) toxicities - Haplotype C3435T, G2677T/A and C1236T: TTT ↑induction death (p: 0.020), ↑renal (p: 0.016) and hepatic (*p* < 0.001) toxicities.
	Rafiee et al., 2019 [63]	942	9.7 (0–30)	Caucasian (81%) Black (13%) Asian (5%) Others (1%)	Yes	De novo	Ara C + IDA + ETOP ± GO	- OS: no influence - EFS: in GO arm CT/TT↑EFS at 5 years (p: 0.017) - DFS: in GO arm CT/TT trend to ↑DFS at 5 years (p: 0.054) - RR: in GO arm CT/TT ↓RR at 5 years (p: 0.003)
	Short et al., 2020 [64]	104	68 (24–88)	Caucasian (86%) Black (13%)	NR	AML 82De novo: 43.9Secondary: 56.1	GO + DAC	- CR, ORR, CIR, OS, RFS: no influence
G1199A rs2229109	Green et al., 2012 [57]	100	63 (20–85)	Caucasian (Europe)	Yes	De novo (normal karyotype)	Ara C + ANT or MIT +/or Others	- OS at 4 years(TX censured): GG suggestive ↓OS (p: 0.06)
	Falk et al., 2014 [59] ^3^	201	59 (18–85)	Caucasian (Sweden)	Yes	De novo (normal karyotype)	Ara C + DAUNO or IDA ± ETOP +/or Others	- CR, OS, EFS: no influence (similar results in FLT3 wild-type subgroup).
C174967T rs6980101	Kim YK et al., 2007 [67]	49	37 (17–69)	Asian (South Korea)	NR	De novo (t (8,21) and inv (16))	Ara C + IDA	- CR: ↑CT vs. CC (p: 0.03) - OS, RFS, RR: no influence
G146792C rs10256836	Kim YK et al., 2007 [67]	49	37 (17–69)	Asian (South Korea)	NR	De novo (t (8,21) and inv (16))	Ara C + IDA	- CR: ↑GG vs. GC (p: 0.03) - OS, RFS, RR: no influence
T134575A rs17327442	Kim YK et al., 2007 [67]	49	37 (17–69)	Asian (South Korea)	NR	De novo (t (8,21) and inv (16))	Ara C + IDA	- CR: ↑TT vs. TA (p: 0.01) - OS, RFS, RR: no influence
A113516G rs4148732	Kim YK et al., 2007 [67]	49	37 (17–69)	Asian (South Korea)	NR	De novo (t (8,21) and inv (16))	Ara C + IDA	- CR: ↑AA vs. AG (p: 0.001) - OS, RFS, RR: no influence
C193T rs121918619	Monzo et al., 2006 [67]	110	44 (16–60)	Caucasian (Spain)	Yes	De novo (intermediate cytogenetic risk)	Ara C + IDA + ETOP	- RR: CC/CT ↑RR (p: 0.02) - OS at 2 years: no influence (but affect in multivariable analysis, CC ↑OS)
Illet144Met	Monzo et al., 2006 [68]	110	44 (16–60)	Caucasian (Spain)	NR	De novo (intermediate cytogenetic risk)	Ara C + IDA + ETOP	- RR, OS: no influence
rs3842 (A>G)	Cao et al., 2017 [20]	206	67.2 (22–98)	Asian (China)	Yes	De novo	Ara C + ANT	- CR: no influence - OS: no influence - RFS: no influence - Toxicity: no influence
rs2235015 (G>T)	Rafiee et al., 2019 [63]	942	9.7 (0–30)	Caucasian (81%) Black (13%) Asian (5%) Others (1%)	Yes	De novo	Ara C + IDA + ETOP ± GO	- OS: no influence - EFS: no influence - DFS: no influence - RR: in GO arm GG/GT ↓RR at 5 years (p: 0.016)
rs2235033 (T>C)	Rafiee et al., 2019 [63]	942	9.7 (0–30)	Caucasian (81%) Black (13%) Asian (5%) Others (1%)	Yes	De novo	Ara C + IDA + ETOP ± GO	- OS: no influence - EFS: no influence - DFS: no influence - RR: no influence
rs1922242 (A>T)	Rafiee et al., 2019 [63]	942	9.7 (0–30)	Caucasian (81%) Black (13%) Asian (5%) Others (1%)	Yes	De novo	Ara C + IDA + ETOP ± GO	- OS: no influence - EFS: no influence - DFS: no influence - RR: no influence
rs1922240 (T>C)	Rafiee et al., 2019 [63]	942	9.7 (0–30)	Caucasian (81%) Black (13%) Asian (5%) Others (1%)	Yes	De novo	Ara C + IDA + ETOP ± GO	- OS: no influence - EFS: no influence - DFS: no influence - RR: no influence
rs1989830 (C>T)	Rafiee et al., 2019 [63]	942	9.7 (0–30)	Caucasian (81%) Black (13%) Asian (5%) Others (1%)	Yes	De novo	Ara C + IDA + ETOP ± GO	- OS: no influence - EFS: no influence - DFS: no influence - RR: no influence
rs2235040 (G>A)	Rafiee et al., 2019 [63]	942	9.7 (0–30)	Caucasian (81%) Black (13%) Asian (5%) Others (1%)	Yes	De novo	Ara C + IDA + ETOP ± GO	- OS: no influence - EFS: no influence - DFS: no influence - RR: no influence
** *ABCB11* **								
rs4668115 (G>A)	Drenberg et al., 2016 [13] ^4^	164	9.1 (0–21)	White (70%) Black (20%) Others (10%)	Yes	De novo	Ara C + DAUNO + ETOP + MIT	- OS: GG ↓OS (p: 0.03) - EFS: GG ↓EFS (p: 0.05)
** *ABCC1* **								
T2684C	Mahjoubi et al., 2008 [82]	111	NR	Arabs (Iran)	NR	52 AMLNR	NR	- CR: no influence - Expression of ABCC1 related to lower CR, drug sensitive and R/R rate
C2007T rs2301666	Mahjoubi et al., 2008 [82]	111	NR	Arabs (Iran)	NR	52 AMLNR	NR	- CR: no influence - Expression of ABCC1 related to lower CR, drug sensitive and R/R rate
G2012T rs45511401	Mahjoubi et al., 2008 [82]	111	NR	Arabs (Iran)	NR	52 AMLNR	NR	- CR: no influence - Expression of ABCC1 related to lower CR, drug sensitive and R/R rate
C2665T	Mahjoubi et al., 2008 [82]	111	NR	Arabs (Iran)	NR	52 AMLNR	NR	- CR: no influence - Expression of ABCC1 related to lower CR, drug sensitive and R/R rate
T825C rs246221	Hampras et al., 2010 [56]	261	61.5 (20–85)	Caucasian (86%) Others (14%) (USA)	Yes	De novo: 75Secondary: 25	Ara C + ANT	- OS (TX censured): no influence - Toxicity: no influence
T1062C rs35587	Hampras et al., 2010 [56]	261	61.5 (20–85)	Caucasian (86%) Others (14%) (USA)	Yes	De novo: 75Secondary: 25	Ara C + ANT	- OS (TX censured): no influence - Toxicity: no influence
G4002A rs2230671	Hampras et al., 2010 [56]	261	61.5 (20–85)	Caucasian (86%) Others (14%) (USA)	Yes	De novo: 75Secondary: 25	Ara C + ANT	- OS (TX censured): no influence - Toxicity: no influence
rs4148350 (G>T)	Megías-Vericat et al., 2017 [62]	225	52.5 (16–78)	Caucasian	Yes	De novo	Ara C + IDA	- CR, induction death: no influence - Toxicity: wild-type GG ↑hepatic severe toxicity grade 3–4 (p: 0.044)
rs129081 (C>G)	Kunadt et al., 2020 [83] ^5^	160	46 (18–60)	Caucasian (Germany)	Yes	NK AMLDe novo: 93.1Secondary: 6.9	Ara C + DAUNO	- CR: no influence - OS: GG↑OS at 5 years (p: 0.035) - DFS: GG↑DFS at 5 years (p: 0.01) - RR: no influence - Toxicity: no influence
rs212090 (A>T)	Cao et al., 2017 [20]	206	67.2 (22–98)	Asian (China)	Yes	De novo	Ara C + ANT	- CR: no influence - OS: no influence - RFS: no influence - Toxicity: AT ↑gastrointestinal toxicity (p: 0.010)
	Kunadt et al., 2020 [83] ^5^	160	46 (18–60)	Caucasian (Germany)	Yes	NK AMLDe novo: 93.1Secondary: 6.9	Ara C + DAUNO	- CR: no influence - OS: no influence - DFS: TT ↓DFS at 5 years (p: 0.021) - RR: no influence - Toxicity: no influence
rs212091 (A>G)	Cao et al., 2017 [20]	206	67.2 (22–98)	Asian (China)	Yes	De novo	Ara C + ANT	- CR: no influence - OS: no influence - RFS: no influence - Toxicity: GG/AG ↓myelosuppression (p: 0.003)
	Kunadt et al., 2020 [83] ^5^	160	46 (18–60)	Caucasian (Germany)	Yes	NK AMLDe novo: 93.1Secondary: 6.9	Ara C + DAUNO	- CR: no influence - OS: GG ↓OS at 5 years (p: 0.006) - DFS: GG ↓DFS at 5 years (p: 0.018) - RR: no influence - Toxicity: no influence
rs3743527 (C>T)	Cao et al., 2017 [20]	206	67.2 (22–98)	Asian (China)	Yes	De novo	Ara C + ANT	- CR: no influence - OS: no influence - RFS: no influence - Toxicity: TT ↑myelosuppression (p: 0.007)
rs4148380 (G>A)	Cao et al., 2017 [20]	206	67.2 (22–98)	Asian (China)	Yes	De novo	Ara C + ANT	- CR: no influence - OS: no influence - RFS: no influence - Toxicity: no influence
** *ABCC2* **								
G4544A rs8187710	Megías-Vericat et al., 2017 [62]	225	52.5 (16–78)	Caucasian	Yes	De novo	Ara C + IDA	- CR, induction death: no influence - Toxicity: no influence
** *ABCC3* **								
45 + 1226 (T>G) rs4148405	Yee et al., 2013 [16] ^6^	154	NR	Caucasian (Europe)	NR	NR	Ara C + ETOP + BUSUL (pre-TX)	- DFS (preTX): GG ↓DFS (p: 9.45 × 10^−6^, remained significant after Bonferroni correction). No influence in non-Caucasian cohort
	Butrym et al., 2021 [85]	95	61 (22–90)	Caucasian (Poland)	Yes	De novo	Ara C + DAUNO or low dose Ara C or AZA	- CR: no influence - OS: G allele ↓OS (p: 0.017)
rs1989983 (G>A)	Yee et al., 2013 [16] ^6^	54	NR	Caucasian (Europe)	NR	NR	Ara C + ETOP + BUSUL (pre-TX)	- DFS (preTX): AA ↓DFS (p: 0.0017). No influence in non-Caucasian cohort
rs2301835 (C>T)	Yee et al., 2013 [16] ^6^	154	NR	Caucasian (Europe)	NR	NR	Ara C + ETOP + BUSUL (pre-TX)	- DFS (preTX): TT ↓DFS (p: 0.0029). No influence in non-Caucasian cohort
rs2277624 (A>G)	Yee et al., 2013 [16] ^6^	154	NR	Caucasian (Europe)	NR	NR	Ara C + ETOP + BUSUL (pre-TX)	- DFS (preTX): AA ↓DFS (p: 0.004). No influence in non-Caucasian cohort
rs8079740 (A>G)	Yee et al., 2013 [16] ^6^	154	NR	Caucasian (Europe)	NR	NR	Ara C + ETOP + BUSUL (pre-TX)	- DFS (preTX): GG ↓DFS (p: 0.0078). No influence in non-Caucasian cohort
rs757420 (T>C)	Yee et al., 2013 [16] ^6^	154	NR	Caucasian (Europe)	NR	NR	Ara C + ETOP + BUSUL (pre-TX)	- DFS (preTX): TT ↓DFS (p: 0.0079). No influence in non-Caucasian cohort
C211T rs4793665	Müller et al., 2008 [18]	139	46.3 (15–86)	Jews (61.2%) Arabs (38.8%)	Yes	De novo	Ara C + ANT ± FLUDA ± MIT	- OS (TX censured): CC ↓OS (p: 0.018)
	Butrym et al., 2021 [88]	95	61 (22–90)	Caucasian (Poland)	Yes	De novo	Ara C + DAUNO or low dose Ara C or AZA	- CR: no influence - OS: no influence
** *ABCG2* **								
G34A rs2231137	Hampras et al., 2010 [56]	261	61.5 (20–85)	Caucasian (86%) Others (14%) (USA)	NR	De novo: 75Secondary: 25	Ara C + ANT	- OS (TX censured): GG ↓OS (p: 0.05) - Toxicity: AA/AG ↑ risk of toxicity grade 3 or more
	Wang et al., 2011 [97]	141	32 (5–70)	Asian (China)	NR	De novoMixed with ALL	Ara C + DAUNO/MITO	- CR: trend to GG ↑CR (p: 0.053). Mixed with ALL patients - OS: GG↑OS (*p* < 0.001). Mixed with ALL patients - DFS: GG↑DFS (*p* < 0.001). Mixed with ALL patients - Haplotype GG (rs2231137) with CA (rs2231142) and CT (rs2231149) ↓DFS/OS (*p* < 0.001)
	Megías-Vericat et al., 2017 [62]	225	52.5 (16–78)	Caucasian	Yes	De novo	Ara C + IDA	- CR, induction death: no influence - Toxicity: no influence
C421A rs2231142	Müller et al., 2008 [18]	139	46.3 (15–86)	Jews (61.2%) Arabs (38.8%)	Yes	De novo	Ara C + ANT ± FLUDA ± MIT	- OS (TX censured): no influence
	Hampras et al., 2010 [56]	261	61.5 (20–85)	Caucasian (86%) Others (14%) (USA)	Yes	De novo: 75Secondary: 25	Ara C + ANT	- OS (TX censured): no influence, but unadjusted HR shown AA ↓OS - Toxicity: no influence
	Wang et al., 2011 [97]	141	32 (5–70)	Asian (China)	NR	De novoMixed with ALL	Ara C + DAUNO/MITO	- CR: no influence. - OS: CC↑OS (*p* < 0.05; lost in multivariate analysis). Mixed with ALL patients - DFS: no influence. Mixed with ALL patients - Haplotype GG (rs2231137) with CA (rs2231142) and CT (rs2231149) ↓DFS/OS (*p* < 0.001)
	Tiribelli et al., 2013 [98]	125	59.2 (20–84)	Caucasian (Italy)	Yes	NR	Ara C + IDA + FLUDA ± ETOP	- OS at 3 years: CC and low ABCG2 expression ↑OS (p: 0.02) - DFS at 3 years: CC and low ABCG2 expression ↑DFS (p: 0.04)
	Megías-Vericat et al., 2017 [62]	225	52.5 (16–78)	Caucasian	Yes	De novo	Ara C + IDA	- CR, induction death: no influence - Toxicity: CA genotype ↑cardiac (p: 0.004) and lung (p: 0.038) toxicities
Ile619Ile (C>T)	Wang et al., 2011 [97]	141	32 (5–70)	Asian (China)	NR	De novoMixed with ALL	Ara C + DAUNO/MITO	- CR, OS, DFS: no influence. Mixed with ALL patients
rs2231149 (C>T)	Wang et al., 2011 [97]	141	32 (5–70)	Asian (China)	NR	De novoMixed with ALL	Ara C + DAUNO/MITO	- CR: no influence. Mixed with ALL patients - OS: CC↑OS (*p* < 0.01; lost in multivariate analysis). Mixed with ALL patients - DFS: CC↑DFS (*p* < 0.05; lost in multivariate analysis). Mixed with ALL patients - Haplotype GG (rs2231137) with CA (rs2231142) and CT (rs2231149) ↓DFS/OS (*p* < 0.001)
rs2231162 (C>T)	Wang et al., 2011 [97]	141	32 (5–70)	Asian (China)	NR	De novoMixed with ALL	Ara C + DAUNO/MITO	- CR, OS, DFS: no influence. Mixed with ALL patients
rs2231164 (C>T)	Wang et al., 2011 [97]	141	32 (5–70)	Asian (China)	NR	De novoMixed with ALL	Ara C + DAUNO/MITO	- CR, OS, DFS: no influence. Mixed with ALL patients

Abbreviations: ALL: acute lymphoblastic leukemia; AML: acute myeloid leukemia; AMSA: amsacrine; ANT: anthracycline; AZA: azacitidine; BH-AC: N4-behenoyl-1D-arabinofuranosycytosine; BUSUL: busulfan; CIR: cumulative incidence of relapse; CR: complete remission; DAC: decitabine; DAUNO: daunorubicin; DFS: disease-free survival; EFS: event-free survival; ETOP: etoposide; FLUDA: fludarabine; GO: gemtuzumab ozogamicin; HWE: Hardy–Weinberg equilibrium; IDA: idarubicin; MIT: mitoxantrone; NK: normal karyotype; NR: not reported; ORR: overall response rate; OS: overall survival; RFS: relapse-free survival; RR: rate of relapse; R/R: relapse/refractory; TX: hematologic transplant. ^1^—Allele frequency and treatment outcomes only reported in 115 patients for C1236T, 142 patients for G2677T/A and 130 patients for C3435T. ^2^—Allele frequency only reported in 103 patients and treatment outcomes only in 44 patients (AML M3 subtype, secondary AML and patients with comorbidities or poor performance status were excluded). ^3^—A total of 100 patients were previously collected and published in Green et al., 2012 [57]. ^4^—This study [13] analyzed 1936 SNPs of 225 genes with a multi-SNP-based approach (including ABC and SLC transporters). Only SNPs with significant results were cited. ^5^—This study [83] included 48 SNPs within 7 genes of 7 *ABC* transporters (*ABCA2*, *ABCA3*, *ABCB1*, *ABCB2*, *ABCB5*, *ABCB7* and *ABCC1*), but only specified the SNPs with significant effect. ^6^—This study [16] analyzed 1659 SNPs of 42 genes with a multi-SNP-based approach. Only SNPs with significant results were cited.

**Table 3 pharmaceutics-14-00878-t003:** Characteristics of the studies included in the systematic review for SNP–SNP combinations of *ABC* and *SLC* transporters.

SNP	Study	n	Age (Range)	Ethnia (Country)	HWE	LMA Status (%)	Chemotherapy Scheme	Clinical Outcomes
** *ABCB1 + SLC* **								
*ABCB1* C3435T rs1045642 *SLCO1B1* rs4149056 (T>C)	Megías-Vericat et al., 2017 [62]	225	52.5 (16–78)	Caucasian	Yes	De novo	Ara C + IDA	- CR, induction death: no influence - Toxicity: TT + TT genotype ↑hepatic toxicity (p: 0.038)
*ABCB1* C3435T rs1045642 *SLC22A16* rs12210538 (A>G)	Megías-Vericat et al., 2017 [62]	225	52.5 (16–78)	Caucasian	Yes	De novo	Ara C + IDA	- CR, induction death: no influence - Toxicity: TT + AA genotype ↑hepatic toxicity (p: 0.019), mucositis (p: 0.004), neutropenia (p: 0.034)
*ABCB1* G2677T/A rs2032582 *SLCO1B1* rs4149056 (T>C)	Megías-Vericat et al., 2017 [62]	225	52.5 (16–78)	Caucasian	Yes	De novo	Ara C + IDA	- CR, induction death: no influence - Toxicity: TT + TT genotype ↑renal (p: 0.030), hepatic toxicity (p: 0.002)
*ABCB1* G2677T/A rs2032582 *SLC22A16* rs12210538 (A>G)	Megías-Vericat et al., 2017 [62]	225	52.5 (16–78)	Caucasian	Yes	De novo	Ara C + IDA	- CR, induction death: no influence - Toxicity: TT + AA genotype ↑hepatic toxicity (p: 0.008)
*ABCB1* G2677T/A rs2032582 *SLC22A16*rs714368 (A>G)	Megías-Vericat et al., 2017 [62]	225	52.5 (16–78)	Caucasian	Yes	De novo	Ara C + IDA	- CR, induction death: no influence - Toxicity: TT + AA genotype ↑renal (p: 0.026), hepatic toxicity (p: 0.008)
*ABCB1* C1236T rs1128503 *SLCO1B1* rs4149056 (T>C)	Megías-Vericat et al., 2017 [62]	225	52.5 (16–78)	Caucasian	Yes	De novo	Ara C + IDA	- CR: no influence - Induction death: TT + TT genotype ↑induction death (p: 0.018) - Toxicity: TT + TT genotype ↑renal (p: 0.048), hepatic toxicity (*p* < 0.001)
*ABCB1* haplotype ^1^ *SLCO1B1* rs4149056 (T>C)	Megías-Vericat et al., 2017 [62]	225	52.5 (16–78)	Caucasian	Yes	De novo	Ara C + IDA	- CR: no influence - Induction death: TT/TT/TT + TT genotype ↑induction death (p: 0.009) - Toxicity: TT/TT/TT + TT genotype ↑renal (p: 0.017), hepatic toxicity (*p* < 0.001)
*ABCB1* haplotype ^1^ *SLC22A16* rs12210538 (A>G)	Megías-Vericat et al., 2017 [62]	225	52.5 (16–78)	Caucasian	Yes	De novo	Ara C + IDA	- CR, induction death: no influence - Toxicity: TT/TT/TT +AA genotype ↑renal (0.036), hepatic toxicity (p: 0.015)
*ABCB1* haplotype ^1^ *SLC22A16* rs714368 (A>G)	Megías-Vericat et al., 2017 [62]	225	52.5 (16–78)	Caucasian	Yes	De novo	Ara C + IDA	- CR, induction death: no influence - Toxicity: TT/TT/TT +AA genotype ↑ hepatic toxicity (p: 0.001)
** *ABCC1 + SLC* **								
*ABCC1* rs4148350 *SLCO1B1/SLC22A16*	Megías-Vericat et al., 2017 [62]	225	52.5 (16–78)	Caucasian	Yes	De novo	Ara C + IDA	- CR, induction death: no influence - Toxicity: no influence
** *ABCC2 + SLC* **								
*ABCC2* rs8187710 *SLCO1B1/SLC22A16*	Megías-Vericat et al., 2017 [62]	225	52.5 (16–78)	Caucasian	Yes	De novo	Ara C + IDA	- CR, induction death: no influence - Toxicity: no influence
** *ABCG2 + SLC* **								
*ABCG2* rs2231142 (C>A) *SLC22A16* rs714368 (A>G)	Megías-Vericat et al., 2017 [62]	225	52.5 (16–78)	Caucasian	Yes	De novo	Ara C + IDA	- CR, induction death: no influence - Toxicity: AC + AA genotype ↑cardiac toxicity (p: 0.033)

Abbreviations: AML: acute myeloid leukemia; CR: complete remission; HWE: Hardy–Weinberg equilibrium; IDA: idarubicin; NR: not reported; OS: overall survival. ^1^—The *ABCB1* haplotype included the polymorphisms rs1128503, rs1045642 and rs2032582.

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
