# Peer review of "Systematic Review of Pharmacogenetics of ABC and SLC Transporter Genes in Acute Myeloid Leukemia"

_pharmaceutics, 2022, doi:10.3390/pharmaceutics14040878_

Round 1

Reviewer 1 Report

The authors systemically reviewed the effects of polymorphism of ABC and SLC transporters on the drug effects and toxicity for acute myeloid leukemia. First of all, the manuscript does not recapitulate the correlation of each SNPs with PK and effect/toxicity. The authors need to be addressed the effect of SNPs on PK of the drug, then should address how the change in PK affects the efficacy and toxicity of the drug. Without this, it is hard for the readers to understand their associations systemically.  Second of all, the manuscript needs to be edited professionally. For instance, some sentences are not correct (e.g., Lines 106-108). Lastly, please delete “Table 29,” as it seems to be irrelevant.  

Author Response

Reviewers Comments to Author:

Reviewer 1:

The authors systemically reviewed the effects of polymorphism of ABC and SLC transporters on the drug effects and toxicity for acute myeloid leukemia. First of all, the manuscript does not recapitulate the correlation of each SNPs with PK and effect/toxicity. The authors need to be addressed the effect of SNPs on PK of the drug, then should address how the change in PK affects the efficacy and toxicity of the drug. Without this, it is hard for the readers to understand their associations systemically. 

#Response to reviewer 1:

Thank you for positive comments. As the reviewer 1 suggested, we have addressed the effect of ABC and SLC SNPs on PK of the drugs employed in AML. Unfortunately studies analyzing together PK and pharmacogenetics are scarce in the field of AML, and it was also explained at the discussion.

OATP1B1 (SLCO1B1)

“The minor allele of rs4149056 has been consistently associated to lower hepatic uptake and higher drug circulating concentrations, increasing the plasma levels and the risk of toxicity in tissues [10,11].” (page 5, at the first paragraph, previously included at manuscript)

SLC22A16

In breast cancer cohorts, variant alleles of SLC22A16 (rs714368) were related to higher exposure levels of doxorubicin and doxorubicinol [6] and dose delays by anthracycline toxicities (lower with rs714368, rs6907567, rs723685 and higher with rs12210538) [7]. (page 5, at the third paragraph, previously included at manuscript)

hENT1 (SLC29A1)

Two nonsynonymous and four synonymous polymorphisms were identified in a functional study of SLC29A1, but no influence in cytarabine uptake was measured [40]. In contrast, the haplotype of three SLC29A1 polymorphisms (−1345C>G, −1050G>A, and 706G>C) was correlated with higher mRNA expression [41]. Another study showed only a modest elevation in hENT1 gene expression with the variant -706G>C, but no influence in cytarabine toxicity in normal blood cells [42].” (page 7, at the first paragraph, marked in red).

P-gp (ABCB1)

“An in vitro study associated the P-gp expression with lower intracellular daunorubicin accumulation [70]. The pharmacokinetics of daunorubicin and his metabolite daunorubicinol were not affected by ABCB1 polymorphisms, as well as mRNA expression in an Indian AML cohort [69]. However, previous studies in breast cancer showed higher doxorubicin clearance and lower peak levels of doxorubicinol with the wild-type haplotype of ABCB1 [47].” (page 7, at the third paragraph, marked in red).

MRP (ABCC1)

“Pharmacokinetic in vitro studies showed with ABCC1 (rs60782127) decreased transport and higher maximum velocity (Vmax) of doxorubicin disposition [80], whereas MRP expression reduced the intracellular daunorubicin accumulation [70].”(page 9, at the second paragraph, marked in red).

BCRP (ABCG2)

“No influence in anthracycline pharmacokinetics was reported with ABCG2 in an AML cohort with daunorubicin (rs2231137, rs2231142, rs769188) [69] and a breast cancer cohort with doxorubicin (rs2231142) [47].” (page 10, at the third paragraph, marked in red).

Discussion

“Influence of ABC pumps in anthracycline pharmacokinetics was suggested by in vitro  [70,80] and studies in other cancers [47], but a population pharmacokinetic study performed in AML failed to reproduce these findings with ABCB1 and ABCG2 polymorphisms [68]. Furthermore, AML studies included did not analyze together the influence of transporters SNPs in drug pharmacokinetic levels and clinical response. In this line, a study in AML demonstrated a correlation between cytarabine plasma level with the CDA genotype, the main enzyme responsible for liver metabolism of cytarabine [104]. In chronic myeloid leukemia, a relevant decrease in imatinib clearance was associated with the variant alleles of ABCB1 and SLCO1B3, [105], as well as in acute lymphoblastic leukemia SLCO1B1 521T>C SNP reduced methotrexate clearance [106]. Previous reviews focused on the impact of ABC and SLC SNPs in drug bioavailability found the same limited evidence of PK studies in the AML context [47,107,108].”.(page 12, at the fifth paragraph, marked in red).

Reviewer 1:

Second of all, the manuscript needs to be edited professionally. For instance, some sentences are not correct (e.g., Lines 106-108).

#Response to reviewer 1:

As the reviewer 1 suggested, a native English speaker has reviewed our manuscript and several grammatical errors, colloquial phrases and syntax changes have been corrected (marked in the text).

Reviewer 1:

Lastly, please delete “Table 29,” as it seems to be irrelevant.

#Response to reviewer 1:

As the reviewer 1 suggested, we have corrected all the edition mistakes at manuscript.

Reviewer 2 Report

This review paper summarizes the pharmacogenetics of ATP binding cassette and solute carriers transporter genes in acute myeloid leukemia. The paper is well organized and the most recent references have been covered. Overall, the paper is in quite good shape and I recommend its publication in the journal of pharmaceutics.

Author Response

Reviewer 2:

Comments to the Author

This review paper summarizes the pharmacogenetics of ATP binding cassette and solute carriers transporter genes in acute myeloid leukemia. The paper is well organized and the most recent references have been covered. Overall, the paper is in quite good shape and I recommend its publication in the journal of pharmaceutics.

#Response to reviewer 2:

Thank you for positive and constructive comments.

Reviewer 3 Report

Recently, a Pinto-Merino et al published a review in Pharmaceutics on the role of pharmacogenetics in the treatment of AML [Pinto-Merino, Á.; Labrador, J.; Zubiaur, P.; Alcaraz, R.; Herrero, M.J.; Montesinos, P.; Abad-Santos, F.; Saiz-Rodríguez, M. Role of Pharmacogenetics in the Treatment of Acute Myeloid Leukemia: Systematic Review and Future Perspectives. Pharmaceutics 2022, 14, 559. https://doi.org/10.3390/ pharmaceutics14030559]. Most of the issues presented in the manuscript are already in the published review, so there is no significant novelty. Thus, in my opinion, this manuscript is not suitable for publication in Pharmaceutics.

Author Response

Reviewer 3:

Comments to the Author

Recently, a Pinto-Merino et al published a review in Pharmaceutics on the role of pharmacogenetics in the treatment of AML [Pinto-Merino, Á.; Labrador, J.; Zubiaur, P.; Alcaraz, R.; Herrero, M.J.; Montesinos, P.; Abad-Santos, F.; Saiz-Rodríguez, M. Role of Pharmacogenetics in the Treatment of Acute Myeloid Leukemia: Systematic Review and Future Perspectives. Pharmaceutics 2022, 14, 559. https://doi.org/10.3390/ pharmaceutics14030559]. Most of the issues presented in the manuscript are already in the published review, so there is no significant novelty. Thus, in my opinion, this manuscript is not suitable for publication in Pharmaceutics.

#Response to reviewer 3:

The cited reference cited by Reviewer 3 was published after our article was submitted (date of submission 25 February 2022; date of Pinto-Merino et al. review was published: 3 March 2022). This review was focused in analyze in general pharmacogenetics in AML, reporting only few studies in AML transporters and two paragraphs at manuscript. However, our review was centred in evaluated the influence of SLC and ABC polymorphisms and their combinations on efficacy and safety in AML cohorts. We opined that both reviewers have enough interest independently, and are focused in different areas of pharmacogenetics. Furthermore, this review was performed following the invitation and the approval of Dr. Mikko Gynther of a brief outline of the review for this special issue.

Round 2

Reviewer 1 Report

The manuscript is improved. 

Author Response

#Response to reviewer 1:

Thank you for positive comments. We do hope you find it interesting and suitable for publication in your journal.

Yours sincerely,

Pau Montesinos (corresponding authors)

Servicio de Hematología y Hemoterapia. Hospital Universitari i Politècnic La Fe

Avda. Fernando Abril Martorell, nº 106

46026 Valencia

Spain

Phone: +34 961245876
